# RT-qPCR Testing and Performance Metrics in the COVID-19 Era

**DOI:** 10.3390/ijms25179326

**Published:** 2024-08-28

**Authors:** Stephen A. Bustin

**Affiliations:** Medical Technology Research Centre, Anglia Ruskin University, Chelmsford CM1 1SQ, UK; stephen.bustin@aru.ac.uk

**Keywords:** RT-qPCR, sensitivity, specificity, diagnostic testing, SARS-CoV-2, COVID-19

## Abstract

The COVID-19 pandemic highlighted the crucial role of diagnostic testing in managing infectious diseases, particularly through the use of reverse transcription-quantitative polymerase chain reaction (RT-qPCR) tests. RT-qPCR has been pivotal in detecting and quantifying viral RNA, enabling the identification and management of SARS-CoV-2 infections. However, despite its widespread use, there remains a notable gap in understanding fundamental diagnostic metrics such as sensitivity and specificity among many scientists and healthcare practitioners. This gap is not merely academic; it has profound implications for interpreting test results, making public health decisions, and affecting patient outcomes. This review aims to clarify the distinctions between laboratory- and field-based metrics in the context of RT-qPCR testing for SARS-CoV-2 and summarise the global efforts that led to the development and optimisation of these tests during the pandemic. It is intended to enhance the understanding of these fundamental concepts among scientists and healthcare professionals who may not be familiar with the nuances of diagnostic test evaluation. Such knowledge is crucial for accurately interpreting test results, making informed public health decisions, and ultimately managing infectious disease outbreaks more effectively.

## 1. Introduction

The polymerase chain reaction (PCR), quantitative PCR (qPCR), and reverse transcription quantitative PCR (RT-qPCR) are arguably the most consequential molecular technologies invented and have had a critical impact on biological, medical, veterinary, and agricultural research [1]. qPCR-based assays can be quantitative, as is generally the case for research applications, or qualitative, as is often the case for diagnostic testing [2]. Unsurprisingly, their speed, sensitivity, and specificity have made qPCR and RT-qPCR the principal molecular diagnostic measuring tools used to establish the identity of specific infectious agents causing infections [3,4,5]. Although highly suited to this task, PCR-based results are affected, like any other tool, by how and under what conditions the technology is applied [6,7,8,9]. This has become especially pertinent following the technology’s ubiquitous use during the COVID-19 pandemic. This has resulted in it being applied and results interpreted by a wide range of users and interested parties, many of which are unfamiliar with its concepts, technical details and strengths and limitations. Whilst this review focuses on the detection of SARS-CoV-2, its concepts and conclusions are generally applicable to all molecular diagnostic testing.

Because RT-qPCR diagnostic testing is used in so many different ways, under different conditions, by different users and in a non-standardised way, the interpretation of test results is somewhat complex. Although the RT-qPCR assay itself is straightforward, the overall testing procedure involves multiple additional pre- and post-test steps that are unrelated to the test itself but have an impact on its apparent performance and interpretation of results [10]. Any one (or more) of these steps can go wrong and make the test appear less reliable than it actually is [11,12,13,14,15,16,17]. 

In general, the RT-qPCR testing process involves the following stages:

Sample collection: The first step is to collect a sample from the person who might be infected with a pathogen. How, by whom, where, and how the sample is collected can make a huge difference to the final test result [18,19]. 

Sample storage, handling, and transport: Different swabs, tubes and solutions used to store a sample affect the stability of the virus and can introduce inhibitors that affect test results. Since many samples are processed at sites remote from where they were taken, the conditions of transport to the laboratory are important. In addition, how a sample is handled throughout this process affects the integrity of viral RNA [20]. 

RNA extraction: In addition to viral proteins and viral RNA, a patient sample contains a mixture of components. This generally requires a procedure to extract and purify the viral RNA. There are many ways of doing this, adding to the potential inconsistencies in results [21].

Reverse transcription: This step converts viral RNA into a DNA template that can be amplified by Taq polymerase. There is a wide range of different reverse transcriptases to choose from, and that choice significantly affects the sensitivity of an RT-qPCR assay [22].

Amplification: The properties of different Taq polymerases also differ, especially with regard to speed of polymerisation, exonuclease activity, thermostability, fidelity and processivity. This makes them the primary source of bias in amplification profiles [23], with additional variability introduced by their preference for certain primer sequence motifs [24].

Detection: The most common method for detecting PCR products is fluorescence-based and, for diagnostic purposes, involves using a PCR amplicon-specific probe, generally a hydrolysis probe or a molecular beacon. It can be DNA-only or have LNA or Pentabase moieties, all of which influence the performance of an assay. Hence, the number of cycles required to amplify the DNA to a detectable level can vary considerably between RT-qPCR tests, even for the same sample [25]. Different RT-qPCR tests can need more or fewer amplification cycles to detect the light emitted by the RT-qPCR amplification product, and other factors can make a difference as well, such as the initial amount of virus in the sample, the characteristics of the virus or which thermal cycler is used [7,26,27,28]. Furthermore, quantification cycle (Cq) values are not a definitive measure as they depend on subjective data analysis and, therefore, should not be used for cut-off settings [29,30,31]. Importantly, altered thresholds alter results, and since the thresholds calculated by instruments from different manufacturers vary depending on what well or combination of wells are being analysed, Cq values also differ [32]. Hence, a Cq value on its own should never be used to compare results obtained with the same or different samples at different times on different instruments using different tests or reagents. Consequently, in terms of practical use of a test, an arbitrary cut-off Cq value to label a target virus as present or absent in a sample is unreliable and should only be used, and then with caution, if cut-offs have been normalised with relative or absolute quantification approaches [33]. This is one reason why many diagnostic tests are qualitative, i.e. record a present/absent result only.

## 2. Diagnostic RT-qPCR

The performance evaluation of diagnostic RT-qPCR assays extends beyond their assessment as research tools and requires additional criteria specific to clinical applications to be considered. Whilst performance metrics obtained during an assay’s development in a controlled laboratory environment are important, they primarily reflect ideal conditions and may not directly translate to real-world diagnostic scenarios. Consistency in diagnostic performance across different testing sites requires an optimised and standardised workflow, a condition that is often not met in practice [34]. Therefore, while laboratory-based metrics provide a foundational understanding of assay performance, the clinically relevant evaluation is determined by metrics that characterise the assay’s effectiveness in accurately diagnosing diseases in individual patients [35]. 

### 2.1. Developmental-Stage Performance Measures

Every diagnostic test, regardless of its intended application, is first developed and optimised in a laboratory where scientists use repeated, tightly controlled measurements to evaluate performance. This involves determining its analytical sensitivity, limit of detection (LoD), specificity, as well as PCR efficiency. These measurements describe the functioning of the test under ideal conditions and do not evaluate its clinical performance.

#### 2.1.1. Limit of Detection or Analytical Sensitivity

During the development phase, the LoD, one of the assessments that define an assay’s “analytical sensitivity” [13], is an important metric that delineates the smallest quantity or concentration of the virus that can be detected consistently by an RT-qPCR assay [36]. LoD is usually assessed by running a large number of viral samples that contain progressively fewer and fewer target molecules under precise lab conditions on a specific instrument to make sure that the assay works correctly and to determine how small a viral sample the test can consistently detect (Figure 1) [37]. It provides a measure of how well the test performs under ideal and strictly monitored conditions. 

However, whilst LoD can be a useful indicator of performance in controlled situations, the test’s ability to identify positive patient samples with different levels of virus can vary quite a bit in real-world scenarios [36], where the testing process involves more steps that are less well controlled, inconsistent sample quality as well as a range of instruments with different characteristics [38,39]. Given these variable factors, once a test has passed and is released into the field, LoD becomes a less meaningful metric [40]. 

One often overlooked factor relates to PCR efficiency, which is critical for both sensitivity and quantification. Efficiency reflects how well the PCR amplifies the target DNA with each cycle, ideally doubling the quantity of DNA (100% efficiency). High efficiency equates to sensitivity, permitting the detection of low amounts of target DNA, whilst inefficient PCR results in failure to detect low-abundant targets and compromises quantification. Therefore, optimising PCR conditions to achieve and maintain high efficiency is essential for both qualitative and quantitative applications. 

A common method for calculating additional sensitivity-associated parameters is to construct a standard curve by plotting Cq values against log-transformed gene copy numbers from accurately quantified control standards. This results in a linear trend line (y = mx + b), where m is the slope and b is the y-intercept. The y-intercept provides a theoretical sensitivity of the assay and describes the number of cycles required for detecting a single target copy. PCR efficiency (E) can be calculated from the slope of the standard curve using the equation E= −1 + 10^(−1/slope)^.

#### 2.1.2. Analytical Specificity 

Analytical specificity measures how good a diagnostic test is at correctly finding the infectious agent it is supposed to detect without getting confused by other pathogens that may be almost the same [13]. There are two parts to this (Figure 2) [3]:

First, a test’s ability to spot different versions of the same infectious agent. With COVID-19, there was the original SARS-CoV-2 virus, followed by a succession of variants such as alpha, delta, many different omicrons, and others. For RNA viruses like SARS-CoV-2, the test’s performance could change as new variants of the virus evolve [41].

Second, a test’s capacity to distinguish the intended target pathogen from other similar ones. For instance, it is crucial to ensure that a test does not identify a different human coronavirus when its single function is to detect SARS-CoV-2. Within a laboratory setting, a test should exhibit absolute specificity, avoiding any instances of false positive outcomes [9].

### 2.2. Real-World Performance Measures

Once a test has moved past the developmental stage and is available for clinical use, additional performance measures come to the fore. These real-world performance assessments are characteristic of diagnostic RT-qPCR tests and lack the precision associated with the development-stage analytical measurements. Instead, they focus on the test’s usefulness in clinical practice—that is, how well the test correctly categorises patient samples as positive or negative for the infectious disease at issue. 

There are four possible outcomes of a diagnostic test (Figure 3):

A positive result, meaning that the target infectious agent is detected:

If the individual was proven to be infected, this is called a true positive (TP). 

If the individual was proven not to be infected, this is called a false positive (FP).

A negative result, meaning that the target infectious agent is not detected:

If the individual was proven not to be infected, this is called a true negative (TN). 

If the individual was proven to be infected, this is called a false negative (FN).

False-negative and false-positive results in RT-qPCR testing for SARS-CoV-2 have been significant challenges during the COVID-19 pandemic. False-negative results are particularly problematic as they may allow infected individuals to unknowingly infect others and can happen for a number of reasons:Early or late in infection, viral load is low, and extracted RNA levels may be below the detection threshold.Improper sample collection technique, storage, and delays in transportation or handling can result in too low an RNA yield or degradation.Suboptimal reagents, inhibitors in the sample or instrument malfunction can compromise the amplification process.

False-positive results occur when a non-infected individual is incorrectly identified as carrying the virus. This is generally caused by cross-contamination during sample collection, handling, or processing, or technical artefacts during the amplification process. False positive results can lead to unnecessary isolation, anxiety, and strain on healthcare resources, diverting attention from truly infected individuals.

Incorrect testing results can be minimised by training personnel in proper swab collection techniques to ensure adequate viral load collection. RNA degradation can be minimised by establishing efficient collection to testing site logistics as well as effecting proper storage and handling of samples. Ensuring physical separation between sample preparation and PCR amplification areas, coupled with decontamination protocols, reduces the risk of contamination and should be standard practice. Assays should ideally detect multiple target regions of the pathogen. It is also essential to incorporate positive controls that contain known viral RNA and, for the detection of RNA viruses, two types of negative controls: one that is RNA-free and another that does not contain an RTase. The former should always be negative; if the latter is positive, then amplification is due to DNA contamination. For ambiguous or low viral load cases, retesting the sample or testing a second sample collected from the patient can provide confirmation, thereby reducing the likelihood of false-negative or false-positive results.

In clinical practice, healthcare providers and infectious disease experts are not concerned about theoretical limits, only whether a test can accurately distinguish between positive and negative samples from real patients. For this purpose, two metrics are most important: (i) the test’s diagnostic sensitivity, defined as the proportion of people with a disease who will have a positive result, and (ii) its diagnostic specificity, defined as the proportion of people without the disease who will have a negative result [42]. These are the performance measures most useful to doctors, independent labs, and health authorities trying to choose which test to use, as they assess the overall accuracy of a test and its ability to correctly classify disease status in a population.

#### 2.2.1. Diagnostic (or Clinical) Sensitivity

Diagnostic sensitivity (D-SN) measures how well a test identifies the disease in real-world situations—i.e., a highly sensitive test has fewer false negative results, allowing patients and providers to feel more confident that a patient who tests negative is, in fact, negative. If the laboratory defined LoD of a diagnostic test were too high, the diagnostic sensitivity statistics would expose this limitation. Specifically, for a test with a high LoD, the sensitivity metric would show that that particular test identified a lower proportion of positive cases compared to an alternative test with a lower LoD. Diagnostic sensitivity is also affected by many other factors that can reduce a test’s accuracy. For example, where the sample was taken from, how it was taken, stored, handled and processed, reagents, equipment and, importantly, the people performing and interpreting the test [17,19,43,44,45,46,47,48]. Because of these outside factors, even a highly sensitive test will sometimes miss finding the infectious agent and return a false negative result. To figure out how well a test is catching true cases, the number of confirmed positive cases is expressed as a percentage of all patients with that condition. D-SN = TP ÷ (TP + FN). This assessment ignores false positives and is, therefore, called the true positive rate (Figure 4).

#### 2.2.2. Diagnostic Specificity

Diagnostic specificity (D-SP) is concerned with a test’s ability to accurately identify individuals who are healthy—that is, a highly specific test has fewer false positive results. Diagnostic specificity is especially important when we want to avoid false alarms or unnecessary worries due to false positive results (Figure 5). 

Just like diagnostic sensitivity, however, diagnostic specificity is affected by various real-world factors that can affect the test’s performance. No test is flawless, and even a highly specific test will sometimes make mistakes or detect contamination, indicating someone has the disease when he does not (a false positive result). To assess a test’s ability to correctly identify true disease-free cases, we look at the number of true negative cases as a percentage of all individuals without the disease. D-SP = TN ÷ (TN + FP). This value, known as the true negative rate, focuses solely on the test’s ability to avoid false alarms and correctly identify non-diseased individuals, hence ignoring false negatives.

#### 2.2.3. Accuracy

Accuracy in a diagnostic context has a very specific meaning: it is defined as the sum of all true positives and true negatives divided by the entire patient population (Accuracy = (TP+TN) ÷ (TP+TN + FP+FN). Whilst diagnostic sensitivity and specificity focus on specific aspects of performance, accuracy offers a more comprehensive view, providing information as to how accurate a test is overall by describing how well it identifies both true positives and true negatives (Figure 6).

However, “accuracy” does not account for false positives or false negatives, such that a test can have high accuracy and still fail to correctly identify many positive patients. To illustrate this point, if in the example shown in Figure 6, the number of true positives had been the same (9 TP), but there had been many more false negatives (20 FN) and true negatives (300 TN), then there would have been many more false negatives (20) than true positives (9). So, whilst the test would have missed more positive cases than it correctly identified, the accuracy of the test would have been much higher at 94%.

#### 2.2.4. Positive and Negative Predictive Values (PPV and NPV)

One problem with these metrics is that whilst they describe how well a diagnostic test correctly identifies true positives or true negatives, they do not account for disease prevalence. This is addressed by the Positive Predictive Value (PPV) and the Negative Predictive Value (NPV) (Figure 7). 

PPV answers the question: What is the probability of someone with a positive test result really having the disease? It considers both true positives and false positives, and a high PPV means that a positive test result is highly indicative of that person actually having the disease. PPV = TP/(TP+ FP).NPV addresses the question: What is the probability of someone with a negative test result truly not having the disease? It considers both true negatives and false negatives, and a high NPV means that a negative test result is highly indicative of that person actually being disease-free. NPV = TN/(TN + FN).

The relative importance of PPV and NPV depends on the prevalence of the condition in the population being tested and the potential consequences of false-positive or false-negative results. A high prevalence of COVID-19 would place more importance on a high PPV, with the aim being to minimise false-positive results and to avoid unnecessary further testing or self-isolation. Low prevalence would place more importance on a high NPV, with the aim being to minimise false-negative results to rule out a condition and minimise the risk of false reassurance.

However, prevalence only provides a snapshot of how common a condition is in a defined population at a single point in time or within a specific time period. Hence, a comparison of PPV and NPV for different tests must use the same populations or different populations with the same incidence of the disease. 

#### 2.2.5. Likelihood Ratios (LR)

The main advantage of LR is their ability to summarise the diagnostic performance of a test in a single value, regardless of disease prevalence. This makes them useful for comparing and interpreting the diagnostic utility of tests across different populations or settings. Clinically, they are useful because they adjust the general pre-test probability to the likelihood of an individual having the disease or not. Likelihood ratios are defined as the percentage of diseased patients with a given test result divided by the percentage of well people with the same test result. 

Positive likelihood ratios (PLRs) are used to measure how much a positive test result changes the odds of a condition being present compared to the odds without the test. It provides information about how well a positive test result can discriminate between individuals with the target condition and those without. PLR = true positive rate/false positive rate, i.e., Sensitivity/(1 − Specificity)Negative likelihood ratios (NLRs) give the change in the odds of having a diagnosis in patients with a negative test. It quantifies how much a negative test result reduces the likelihood of the target condition being present compared to the likelihood without the test result. NLR = false negative rate/true negative rate, i.e., (1 − Sensitivity)/Specificity

#### 2.2.6. Confidence Intervals

Confidence intervals (CI) reflect the inherent uncertainty associated with how even the best test can give dissimilar results in different situations. This is, of course, especially relevant when considering the variability that is generally associated with real-world data. Since the confidence range changes with more performance data, a diagnostic test’s CI depends on the number of available data points. The more consistent the test is over a larger number of data points, the narrower the range of possible outcomes, and the more confident we are in the reported performance measures. In practice, the CI range is determined by stating a particular level of certainty, with diagnostic tests commonly reporting their performance metrics using a 95% CI. 

## 3. Limitations of RT-qPCR Testing and Performance Measures

It is widely recognised that no diagnostic test can achieve 100% diagnostic sensitivity or specificity in all circumstances. Indeed, this has been stated many times for RT-qPCR tests well before the COVID-19 pandemic [49,50,51,52,53,54,55,56]. With regards to RNA viruses, previous analyses of tests targeting the novel coronaviruses SARS-CoV [57] and MERS-CoV [58], as well as other respiratory viruses [59,60], clearly showed that whilst RT-qPCR tests were highly sensitive and specific, none reached 100% in every evaluation 100% of the time.

In the context of SARS-CoV-2 and COVID-19, it was (and still is) especially implausible that an RT-qPCR test could be 100% sensitive or specific in every evaluation [61]. Due to the novel and uncertain nature of the virus, it was unknown how the disease would manifest across different individuals and populations [62], with numerous other factors, such as host immune response and site of specimen collection, affecting test performance [63,64,65]. Even today, there is no diagnostic test that reliably determines the presence of COVID-19 in every situation all of the time [66]. 

The challenges of diagnostic test development and the emphasis on maintaining realistic expectations regarding sensitivity and specificity are nothing new and have been discussed for decades [67,68,69,70]. Consequently, the inherent variability in test performance has long been widely acknowledged, and this knowledge informed the expectations surrounding COVID-19 diagnostic tests [71]. It was clear that RT-qPCR tests had variable diagnostic sensitivity and specificity and that test results needed to be evaluated as part of a multiplexed, clinically integrated diagnostics approach [72,73,74]. Indeed, it was well-known back in 2020 that sampling site and time of testing are key determinants of whether SARS-CoV-2 infected individuals are identified by RT-qPCR [75]. This, coupled with media-wide dissemination of expert opinions and research findings, helped build public awareness about the complexities of diagnostic testing and the need for a nuanced interpretation of test results.

Usually, in evaluating an RT-qPCR test, an accurate evaluation of performance (e.g., diagnostic sensitivity and specificity) requires definitive knowledge of an individual’s infection status. However, such certainty is unattainable with respect to a new virus for which no established “gold standard” test yet exists [76,77]. In the absence of this gold standard, carefully evaluated comparative testing of a diagnostic RT-qPCR test against others can be a useful means of estimating their real-life performance [78]. Typically, this involves selecting a reference test that has been demonstrated to be reliable and has been validated for the detection of the virus in relevant populations. The different tests are then used under the same conditions, in the same laboratory, and at the same time to amplify the same, previously characterised samples. The sensitivity and specificity of the various tests can then be assessed by comparing the results of each sample and calculating the proportion of true positives, false positives, true negatives, and false negatives when compared to the reference test. 

This comparative approach was commonly employed early in the COVID-19 pandemic and allowed scientists to estimate an RT-qPCR test’s sensitivity and specificity even in the absence of complete information on the infection status of all samples tested. However, this estimation relied heavily on the reliability of the chosen reference test: if the reference test was less sensitive or specific than the tests it was being compared against, then the accuracy of the assessment would be compromised. In other words, the reference test must not be less sensitive or specific than the tests that are being compared to it. If an inaccurately verified status is used as the benchmark, the diagnostic accuracy estimates will be biased [79,80].

## 4. RT-qPCR Testing in the Context of the COVID-19 Pandemic

The release of the genomic sequence of the novel coronavirus SARS-CoV-2 on January 10, 2020 [81] allowed for the development of molecular diagnostic kits for its detection. Despite the unprecedented lack of understanding of the nature of this pandemic [82], the complexities involved in diagnostic test development and evaluation [83,84] and the short time frame in which to achieve these, the scientific community reacted with unprecedented speed to produce a number of RT-qPCR tests for SARS-CoV-2 [61]. Since there was little information about mutation, substitution, or recombination rates of the virus [85,86], these tests targeted a range of viral genes, with some tests detecting single genes and others multiple ones. Unsurprisingly, differences in assay designs, reagents, protocols, and equipment resulted in differences in the performance of these tests [38,87,88,89,90,91]. Predictably, as discussed earlier, it was also not a surprise that the performance of the different test kits varied with the populations being tested, their disease status, the quality of the sample collected, extraction procedures as well as analysis methods [17,19,43,44,45,46,47,48]. Nevertheless, independent evaluations of many of these tests showed that the analytical sensitivities and amplification efficiencies of most tests were sufficient for their use as diagnostic tests [92,93,94]. Crucially, the landscape of COVID-19 testing was constantly evolving, and any evaluation of testing performance was subject to change as more real-life data became available [4]. 

As a consequence, despite testing programmes themselves being wholly inadequate, poorly organised and surrounded by confusion and misinformation [95], the early RT-qPCR tests for SARS-CoV-2, whilst not perfect, succeeded in achieving their main purpose within the limitations of the circumstances: accurate and fast detection of the virus in infected individuals [4]. As time moved on and testing became more widespread, improved sampling protocols, more experienced laboratory staff, and more judicious interpretation of test results led to an overall improvement in the real-world performance and reliability of COVID-19 RT-qPCR tests even when the test assays remained the same. These efforts were instrumental in mitigating the pandemic’s impact and guiding public health responses.

## 5. Conclusions

In summary, the RT-qPCR test has been instrumental in helping manage the COVID-19 pandemic by providing a reliable method for detecting and quantifying SARS-CoV-2 RNA. However, the complexity of the RT-qPCR testing process and the multitude of factors influencing its outcomes necessitate a comprehensive understanding of both laboratory- and field-based performance metrics. The distinction between analytical sensitivity, specificity, and their real-world diagnostic counterparts—sensitivity and specificity—highlights the challenges and limitations faced when interpreting test results in clinical settings.

Developmental-stage performance measures, such as the limit of detection and analytical specificity, provide crucial insights into a test’s capabilities under controlled conditions. These metrics, however, may not fully translate to real-world scenarios where sample handling, transport conditions, and variability in testing environments have a significant impact on the accuracy of diagnostic outcomes. Consequently, real-world performance measures, including diagnostic sensitivity, specificity, accuracy, and predictive values, become critical for assessing the practical utility of RT-qPCR tests in identifying SARS-CoV-2 infections accurately.

The inclusion of likelihood ratios and confidence intervals further refines our understanding of a test’s diagnostic performance, offering a more nuanced interpretation that accommodates varying disease prevalence and the inherent uncertainty in test results. Despite these sophisticated metrics, the RT-qPCR test, like any diagnostic tool, is not infallible. It is essential for scientists and healthcare professionals to remain aware of the limitations and potential sources of error that can affect test outcomes. This awareness fosters a more critical and informed application of RT-qPCR tests in managing current and future infectious disease outbreaks. Ultimately, bridging the knowledge gap regarding fundamental diagnostic metrics will enhance the effective use of RT-qPCR testing, contributing to improved patient outcomes and more robust public health strategies.

## Figures and Tables

**Figure 1 ijms-25-09326-f001:**
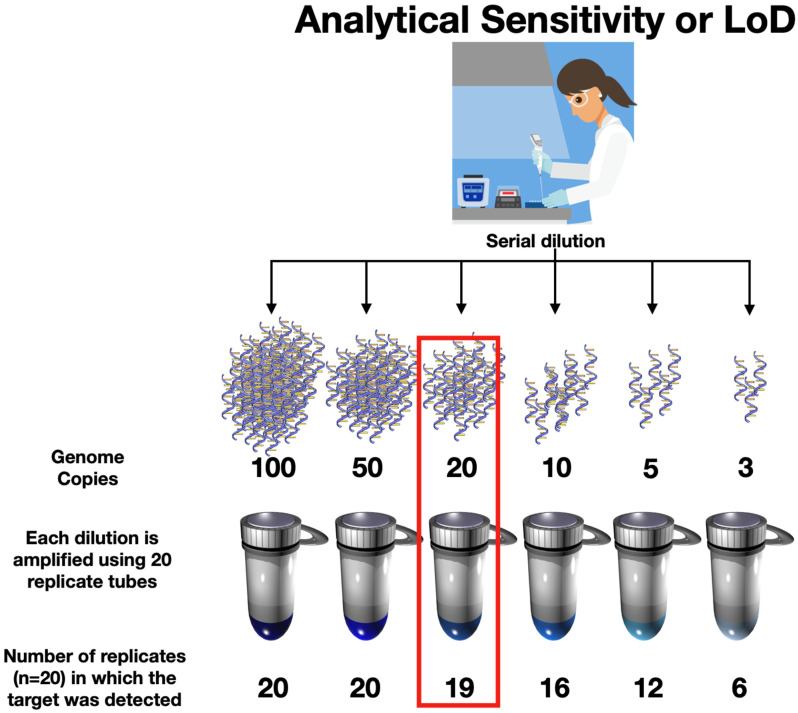
The LoD for an RT-qPCR test is typically established by employing meticulously quantified purified nucleic acid samples, experienced laboratory workers, rigorously defined conditions, and specific instruments. It involves using samples where the precise count of viral genomic RNA copies per microliter (µL) has been established. In this example, that count spans from 100 copies down to 3 copies per µL. Each of these samples is subjected to 20 repeated RT-qPCR reactions, and the number of reactions with positive outcomes and those remaining blank are counted. Depending on the chosen confidence threshold, the LoD is subsequently computed. In this example, the lowest concentration of template that produced at least 95% positive replicates is 20.

**Figure 2 ijms-25-09326-f002:**
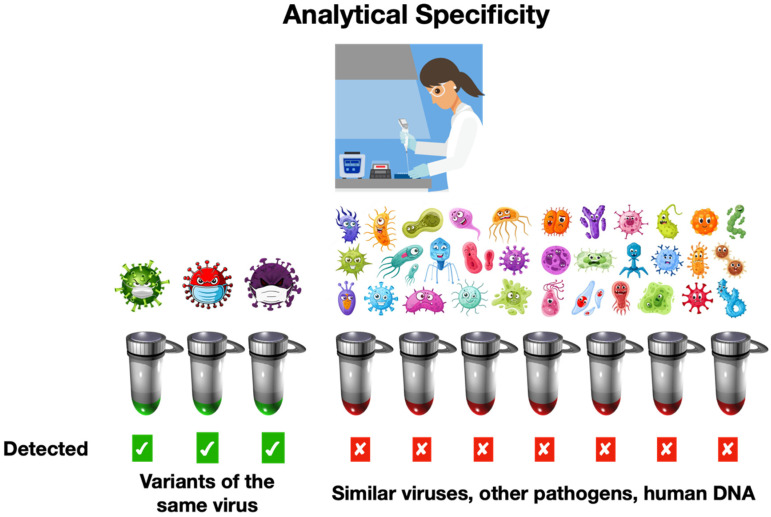
Analytical specificity for an RT-qPCR test is typically established in a laboratory under tightly controlled conditions by screening hundreds of different genomic RNA and DNA samples purified from as wide a range of infectious agents as possible. **A.** A specific test will detect all variants of the same pathogen (e.g., SARS-CoV-2 alpha, delta, omicron). **B.** It will not detect similar viruses such as SARS-CoV and MERS, as well as human coronaviruses and any other of the numerous viral, bacterial, fungal, and protozoan pathogens. It will also not detect human genomic RNA and DNA samples. A diagnostic test is expected to be 100% specific for the targeted pathogen and not detect any of the related or unrelated pathogens.

**Figure 3 ijms-25-09326-f003:**
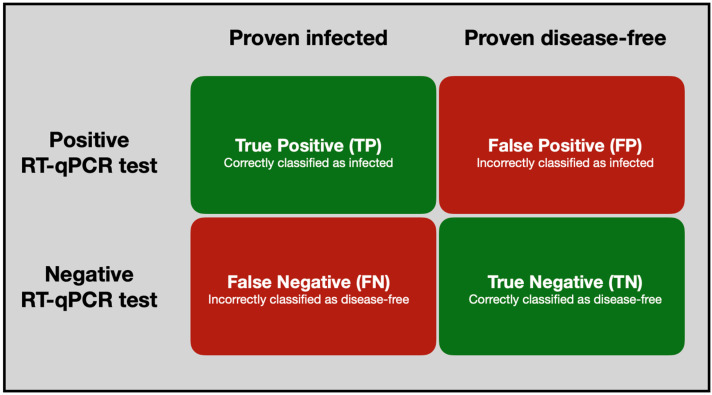
Punnett square for RT-qPCR test sensitivity and specificity.

**Figure 4 ijms-25-09326-f004:**
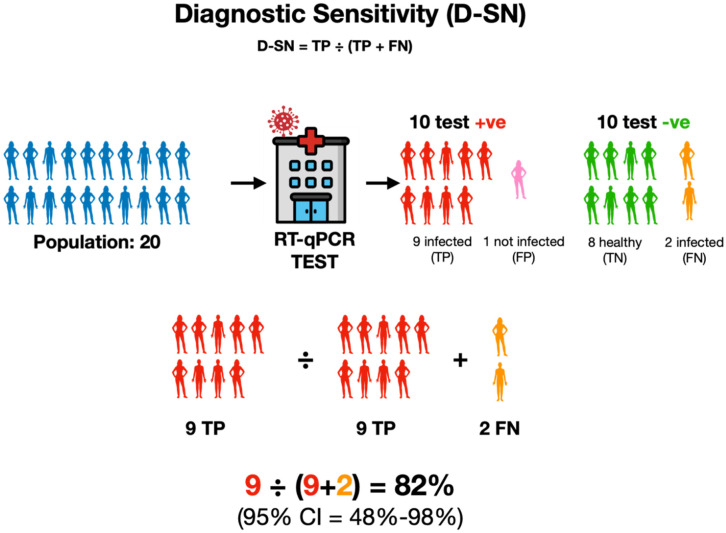
Diagnostic sensitivity emphasises the ability of a test to catch all actual positives without considering any false-positive results. It is expressed as the percentage of true-positive tests out of all individuals who are known to be infected, regardless of whether they tested positive (TP) or not (FN). In this example, in a population of 20 individuals shown in blue, eleven were infected, but only nine tested positive by RT-qPCR (+ve, red, TP), whereas two tested negative (-ve, orange, FN). One of the nine patients testing positive was healthy (pink, FP). The sensitivity is therefore calculated by dividing the number of true positive tests (9) by the total number of tests that should have been positive (11).

**Figure 5 ijms-25-09326-f005:**
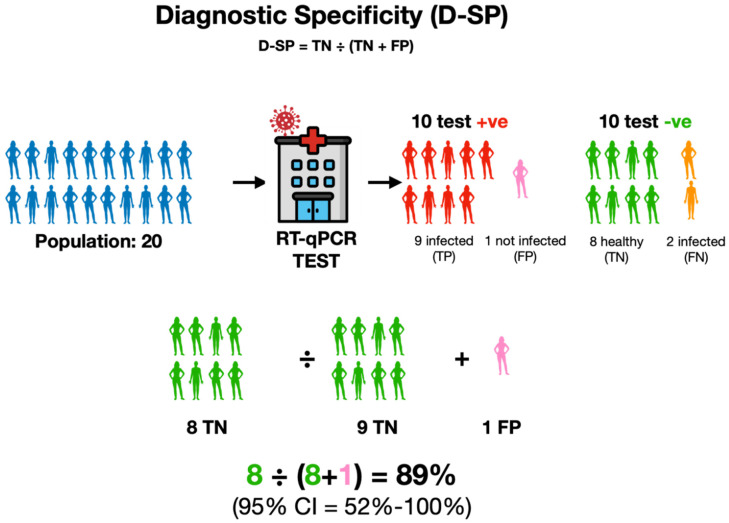
Diagnostic specificity measures the ability of the test to correctly detect all actual negatives without considering any false negatives. It is expressed as the percentage of all true-negative tests out of all individuals who are known not to be infected, regardless of whether they tested negative (TN) or not (FP). In this example, in a population of 20 individuals shown in blue, nine individuals were healthy, but only eight tested RT-PCR negative (-ve, green, TN), whereas one tested positive (pink, FP). The specificity is therefore calculated by dividing the number of true positive tests (8) by the total number of tests that should have been negative (9).

**Figure 6 ijms-25-09326-f006:**
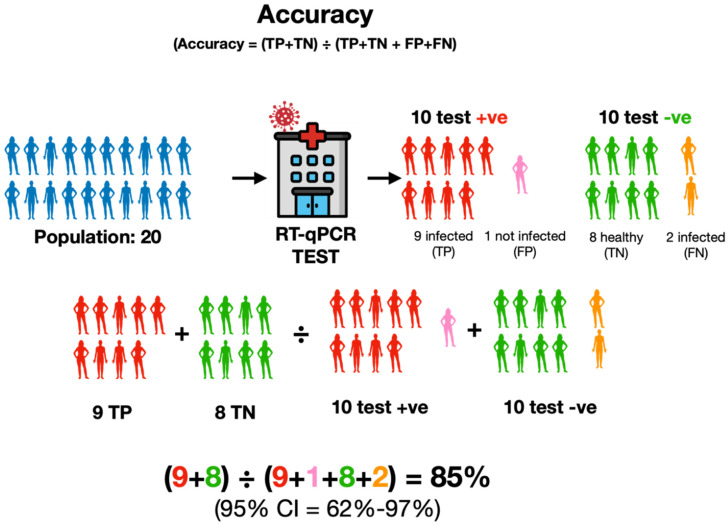
Accuracy is a measure of how well the test identifies both TP and TN and does not consider FP and FN. It is expressed as the percentage of all true-positive and all true-negative tests out of all individuals that have been tested. In this example, in a population of 20 individuals shown in blue, there were nine true-positive (TP, red) and eight true-negative (TN, green) individuals. Since 20 individuals were tested, accuracy is calculated by dividing the 17 individuals with a correct test by all 20 individuals tested, even though there are one FP (pink) and two FN (orange) amongst them.

**Figure 7 ijms-25-09326-f007:**
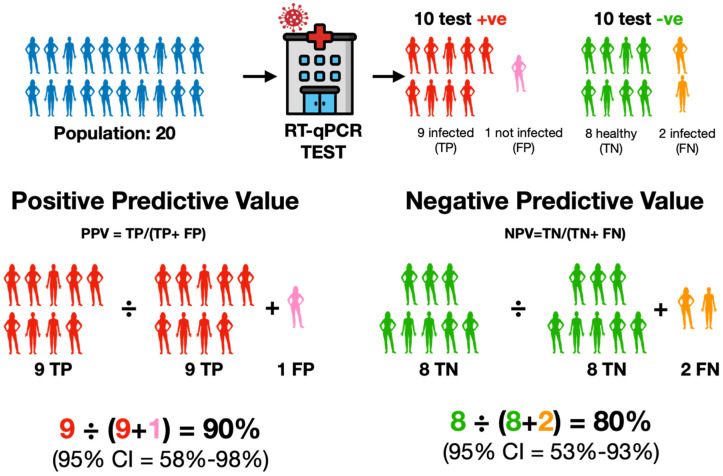
PPV specifically emphasises the ratio of true positives (TP, red) within all cases that the test identified as positive (TP + false positives, FP, pink). It addresses the precision of the test’s positive outcomes among all those that have been indicated as positive by the test. NPV specifically emphasises the ratio of true negatives (TN, green) within all cases that the test identified as negative (TN + false negatives, FN, orange). It addresses the precision of the test’s negative outcomes among all those that have been indicated as negative by the test. Hence, PPV characterises the reliability of a positive result, while NPV characterises the reliability of a negative result.

## Data Availability

Not applicable.

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
