# Peer review of "RT-qPCR Testing and Performance Metrics in the COVID-19 Era"

_ijms, 2024, doi:10.3390/ijms25179326_

Round 1
Reviewer 1 Report
Comments and Suggestions for Authors
RT-qPCR testing and performance metrics in the COVID-19 era
The present review article by the authors aims to provide knowledge of the RT-qPCR to fill the gaps in understanding fundamental diagnostic metrics such as sensitivity and specificity among scientists and healthcare practitioners. This review clarifies the distinctions between laboratory- and field-based metrics in RT-qPCR testing for SARS-CoV-2. It summarizes the global efforts that led to developing and optimizing RT-qPCR testing during the pandemic. The review was well-formulated and perfectly organized and may be helpful for public health decision-making. Following are the specific comments to further strengthen the manuscript,
1. LoD, Given these variable factors, once a test is past the development phase, LoD becomes a less meaningful metric. Do the authors want to say once the test is passed?
2. Is the color coding in Figure 2 appropriate?
3. Expand more about false negative and false positive RT-qPCR result outcomes and discuss how this can be controlled in field testing for SARS-CoV-2 in the context of the COVID-19 pandemic.
Author Response
- LoD, Given these variable factors, once a test is past the development phase, LoD becomes a less meaningful metric. Do the authors want to say once the test is passed?
Text has been changed
- Is the color coding in Figure 2 appropriate?
As reviewer 2 pointed out the same issue, this has been changed.
- Expand more about false negative and false positive RT-qPCR result outcomes and discuss how this can be controlled in field testing for SARS-CoV-2 in the context of the COVID-19 pandemic.
This has been done
All changes are highlighted in the resubmitted manuscript
Reviewer 2 Report
Comments and Suggestions for Authors
The review ‘RT-qPCR testing and performance metrics in the COVID-19 era’ try to clarify the distinctions between laboratory and field-based metrics in the context of RT-qPCR testing for SARS-CoV-2. The manuscript is very interesting but there are some issues that need to be addressed:
Before 'This can make the interpretation of RT-qPCR diagnostic test results somewhat complicated' (page 2), it is necessary to add one or more sentences that connects the previous period with the rest of the review, where only the RT-qPCR of Sars COV (RNA virus...) is mentioned.
For the paragraphs: Reverse transcription, Amplification, Detection, a short description must be added as was done for the other phases (Sample collection …).
This period ‘whilst the measurements applied to a test whilst still being developed in the lab are informative and important, they are correct only in the controlled laboratory environment and have less relevance to real-life diagnostic situations. This is because consistency in the field can only be achieved if there is an optimised and, crucially, standard test workflow setup across testing sites (34), which in practice is never the case. is unclear’ (page 4) can it be rewritten?
Both 'Developmental-Stage Performance Measures’ and 'Real-World Performance Measures', it would be better to organize them into paragraphs and sub-paragraphs in order to make reading easier (for example- 1. 'Developmental-Stage Performance Measures, 1.1 LoD, 1.2 Analytical Specifity).
Use in both cases either ‘or’ or ‘and’ (analytical sensitivity, or limit of detection (LoD)/ LoD and Analytical Sensitivity), or better explain the difference between LoD and Analytical Sensitivity.
In the sentence ‘This involves determining its analytical sensitivity, or limit of detection (LoD), specificity, PCR efficiency and’ are words missing?
At the end of this sentence (It provides a measure of how well the test performs under ideal and strictly monitored conditions.) it should be added how sensitivity is calculated, as done in the figure 1.
In the sentence ‘This involves determining its analytical sensitivity, or limit of detection (LoD), specificity, PCR efficiency and’, the author indicates: analytical sensitivity, or limit of detection (LoD), specificity, PCR efficiency, but later he dedicates 2 subparagraphs only to analytical sensitivity, or limit of detection (LoD) and specificity, but he doesn't write anything about PCR efficiency, why?
To make this period easier to understand (page 8- There are four possible outcomes of a diagnostic test:• A positive result, meaning that the target infectious agent is detect…) is it possible to add a summary table (positive to test and infection (+/+) TP, positive to test but negative to infection (+/-) FP) like Punnet square?
It would be more intuitive to write D-SN and D-SP instead of SN and SP as an acronym for Diagnostic Sensitivity and Diagnostic Specificity(pages 9 and 10).
Page 15: it is more correct to write MERS-CoV than just MERS, since we are talking about the virus as SARS-CoV (as well as in Figure 2).
Figure 1: Given that in this figure we are talking about RT-qPCR for Sars-CoV 2, would it be more correct since it is an RNA virus to put a single-stranded molecule as genome copies?
Figure 1: Since they are dilutions of the same type of sample, it would be clearer that the liquid inside the tubes are all in the darkest to lightest shade of blue.
Figure 2: Invert the colors green (identified right) with red (unidentified).
Figure 2: Separate the 2 parts: A) identify variants of the same virus B) do not identify similar viruses or other pathogens.
Figure 5 in the page 11: change number in figure 4 (4 no 5).
Figure 3, 4 (ex 5), 5 and 6: all 4 figures initially describe the same thing: 10 positive tests of which 9 are infected (TP) and one is not infected (FP), and 10 negative of which 8 are not infected (TN) and 2 infected (FN). Rewrite the text of the figures well so that it is clearer and more equal in all 3 I don't know what +ve and -ve mean.
Figure 3,4,5 and 6: Adding the general Formula to each figure.
Figure 5 (page 12): it would be more correct to write (9+8/9+8+1+2).
Watch out for minor typos and graphics errors.
I hope I have been of help to you.
Author Response
Before 'This can make the interpretation of RT-qPCR diagnostic test results somewhat complicated' (page 2), it is necessary to add one or more sentences that connects the previous period with the rest of the review, where only the RT-qPCR of Sars COV (RNA virus...) is mentioned.
This has been done
For the paragraphs: Reverse transcription, Amplification, Detection, a short description must be added as was done for the other phases (Sample collection …).
This has been done
This period ‘whilst the measurements applied to a test whilst still being developed in the lab are informative and important, they are correct only in the controlled laboratory environment and have less relevance to real-life diagnostic situations. This is because consistency in the field can only be achieved if there is an optimised and, crucially, standard test workflow setup across testing sites (34), which in practice is never the case. is unclear’ (page 4) can it be rewritten?
This has been done
Both 'Developmental-Stage Performance Measures’ and 'Real-World Performance Measures', it would be better to organize them into paragraphs and sub-paragraphs in order to make reading easier (for example- 1. 'Developmental-Stage Performance Measures, 1.1 LoD, 1.2 Analytical Specifity).
This has been implemented
Use in both cases either ‘or’ or ‘and’ (analytical sensitivity, or limit of detection (LoD)/ LoD and Analytical Sensitivity), or better explain the difference between LoD and Analytical Sensitivity.
This has been implemented
In the sentence ‘This involves determining its analytical sensitivity, or limit of detection (LoD), specificity, PCR efficiency and’ are words missing?
This involves determining its analytical sensitivity, or limit of detection (LoD), specificity as well as PCR efficiency.
At the end of this sentence (It provides a measure of how well the test performs under ideal and strictly monitored conditions.) it should be added how sensitivity is calculated, as done in the figure 1.
This has been done
In the sentence ‘This involves determining its analytical sensitivity, or limit of detection (LoD), specificity, PCR efficiency and’, the author indicates: analytical sensitivity, or limit of detection (LoD), specificity, PCR efficiency, but later he dedicates 2 subparagraphs only to analytical sensitivity, or limit of detection (LoD) and specificity, but he doesn't write anything about PCR efficiency, why?
This has been done
To make this period easier to understand (page 8- There are four possible outcomes of a diagnostic test:• A positive result, meaning that the target infectious agent is detect…) is it possible to add a summary table (positive to test and infection (+/+) TP, positive to test but negative to infection (+/-) FP) like Punnet square?
This has been implemented
It would be more intuitive to write D-SN and D-SP instead of SN and SP as an acronym for Diagnostic Sensitivity and Diagnostic Specificity(pages 9 and 10).
This has been implemented
Page 15: it is more correct to write MERS-CoV than just MERS, since we are talking about the virus as SARS-CoV (as well as in Figure 2).
This has been implemented
Figure 1: Given that in this figure we are talking about RT-qPCR for Sars-CoV 2, would it be more correct since it is an RNA virus to put a single-stranded molecule as genome copies?
This has been changed
Figure 1: Since they are dilutions of the same type of sample, it would be clearer that the liquid inside the tubes are all in the darkest to lightest shade of blue.
This has been changed
Figure 2: Invert the colors green (identified right) with red (unidentified).
This has been changed
Figure 2: Separate the 2 parts: A) identify variants of the same virus B) do not identify similar viruses or other pathogens.
This has been changed
Figure 5 in the page 11: change number in figure 4 (4 no 5).
The text has been updated due to new Figure 4. Other Figures also adjusted,
Figure 3, 4 (ex 5), 5 and 6: all 4 figures initially describe the same thing: 10 positive tests of which 9 are infected (TP) and one is not infected (FP), and 10 negative of which 8 are not infected (TN) and 2 infected (FN). Rewrite the text of the figures well so that it is clearer and more equal in all 3 I don't know what +ve and -ve mean.
This has been changed
Figure 3,4,5 and 6: Adding the general Formula to each figure.
This has been changed
Figure 5 (page 12): it would be more correct to write (9+8/9+8+1+2).
This has been changed
Watch out for minor typos and graphics errors
This has been changed
Round 2
Reviewer 2 Report
Comments and Suggestions for Authors
I hope I help you for a good work!